# Impact of Inflation in Different States of Unemployment: Evidence with the Phillips Curve in South Africa from 2008 to 2022

Eugene Msizi Buthelezi

Department of Economics, University of Free State, Bloemfontein 9301, South Africa; msizi1106@mail.com or butheleziem@ufs.ac.za

**Abstract:** This paper investigates the impact of inflation in different states of unemployment: evidence with the Phillips curve in South Africa. The contribution of this paper is to examine the impact of inflation on different states of unemployment in South Africa. The Paper employs Markov-switching dynamic regression and data from 2008 to 2022. It was found that there are 2 states of unemployment mean rates of 25.55% and 33.59%, expected to run for 67 and 7 quarters, respectively. There is a 98.51% and 86.99% chance of a move and then returning to the same state, respectively. In states 1 and 2, a 1% increase in inflation results in a 2.61% increase and a 0.06% decrease in unemployment, respectively. There are times when the Phillips curve rationale is not holding. The government needs to increase the channels of employment opportunities. There is a need to re-look at the trade-offs between inflation and unemployment in the economy.

**Keywords:** Phillips's curve; inflation; unemployment; Markov-switching dynamic regression (MSDR)

## 1. Background

The relationship between unemployment and inflation has been at the centre of macroeconomics. However, no consensus has been reached on the theoretical and empirical bases (Amusa et al. 2013; Okafor et al. 2016; Phillips 1958; Roberts 1995; Vermeulen 2017). Theoretical Phillips (1958) advocated for a negative relationship between unemployment and inflation. On the other hand, Roberts (1995) argues that the Phillips Curve, which accounts for the sticky-inflation model, may be limited to account for rational expectations. This indicates that it exhibits less data adjustment due to the supposedly imperfect rationality of the agents as to their expectation formation change. Ball and Mankiw (1995) note that there is an erroneous interpretation that reflects the unproven link between inflation and unemployment as a causal relationship. McCallum (1997), on the other hand, Phillips (1958) influences policy, does not take into account other drivers of inflation and attributes the reason for inflation to growth. Empirically, there is no agreement on the relationship between unemployment and inflation. Scholars who have found a negative relationship between unemployment and inflation include Hodge (2002), Al-zeaud and Al-hosban (2015), Vermeulen (2015), and Okafor et al. (2016), among others. Scholars that have found a positive relationship between unemployment and inflation include Bhattarai (2016) and Vermeulen (2017), among others. Scholars that have found no clear relationship between unemployment and inflation include Rocheteau et al. (2007) and Asif (2013), among others.

In the context of South Africa, fiscal authorities take unemployment very seriously, and it is included in the top 3 economic challenges, including poverty and inequality. South Africa has been characterized by unemployment rates of 26.53% as well as an inflation rate of 5.59%[1] between 2008 quarter 1 and 2022 quarter 1 (SARB 2022a). There have been interventions by policymakers over the years to reduce the level of unemployment and inflation. On the fiscal policy side, the National Development Plan (NDP) outlines the objective of the government to create 11 million jobs in South Africa by 2030 (NDP 2013).

Moreover, the policy outlines a target of reducing unemployment by 14% in 2020 and 6% in 2030 (NDP 2013). In 2022, unemployment is 12.53% higher than the target of 2020. In addition, SA recorded an unemployment rate of 34.50% in 2022 quarter 1 (SARB 2022a). On the other hand, at a monetary policy level in the year 2000, the South Africa Reserve Bank (SARB) adopted an inflation-targeting (IT) monetary policy in the range of 3% to 6%. The inflation targeting policy was adopted to ensure price stability (SARB 2022a). Since 2008, two economic events have occurred: the global financial crisis and COVID-19. To stabilize the price, the SARB reduced the repurchase rate by 650 basis points between 2008 and 2010 and cut the repurchase rate by 225 basis points during the period of COVID-19. The backing of IT is that SA can attain low inflation and improve policy credibility as well as investor confidence, among other things (SARB 2022b). Nonetheless, despite these interventions, the inflation rate average from 2008 to 2022 is 5.58%, which is at the higher bound.

Proponents of the hypothesis of the Phillips curve advocate that there is a trade-off between inflation and unemployment, which might be exploited to reduce the unemployment rate Phillips (1958). Figure 1, graph a, reflects a negative relationship between unemployment and inflation, which shows that there is an instance where the rationale of the Phillips curve may hold. On the other hand, graph b reflects that there is no positive or negative relationship between inflation and unemployment. Graph c reflects that there is a positive relationship between inflation and unemployment in South Africa. Given this inconsistency, it is important to investigate the impact of inflation in different states of unemployment. This needs to be carried out to understand the impact of inflation at a high level of unemployment as well as a lower unemployment level.

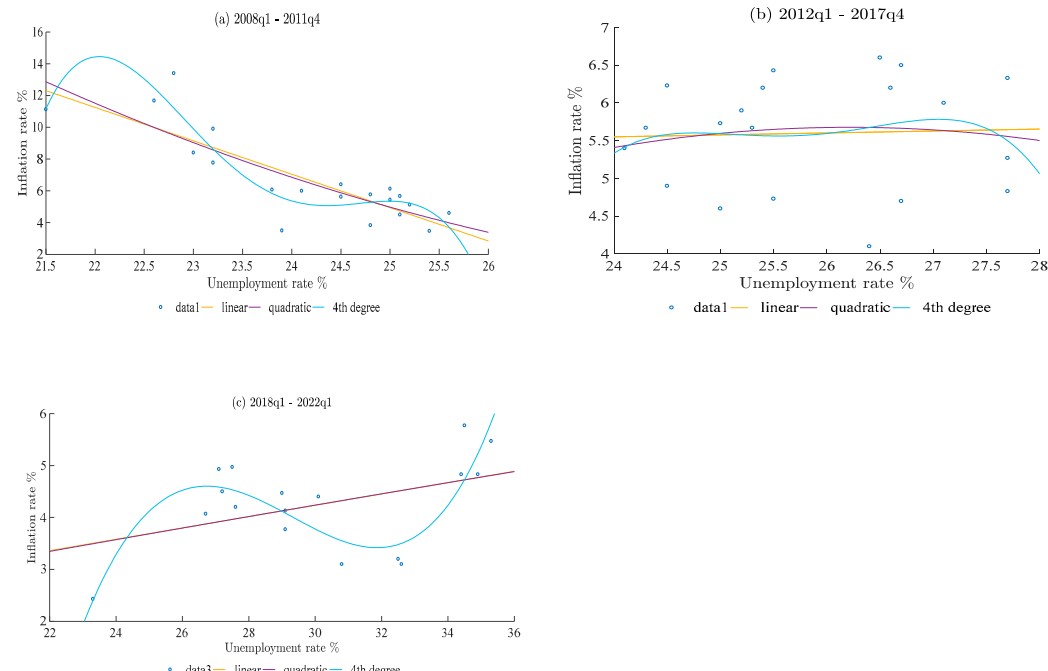

**Figure 1.** Inflation and unemployment in South Africa. The graphs are done by the author.

Given the high level of unemployment as well as inflation in South Africa. It is critical to investigate the effect of inflation on unemployment. The key questions of this paper are as follows: what is the impact of inflation in different states of unemployment? How long will the unemployment rate be at a higher rate of unemployment and a lower rate of unemployment? What is the probability of transition to different states of unemployment? Given these questions, the purpose of this paper is to investigate the impact of inflation in different states of unemployment with evidence of the Phillips curve in South Africa. The contribution of this paper is to examine the impact of inflation in different states of

unemployment. The paper employs Markov-switching dynamic regression (MSDR) using time-series data from 2008 quarter 1 to 2022 quarter 1. It was found that there are 2 states of unemployment with mean rates of 25.55% and 33.59%. These states are expected to run for 67 and 7 quarters, respectively. Moreover, there is a 98.51% and 86.99% chance of a move and then returning to the same state in both states 1 and 2, respectively. In states 1 and 2, a 1% increase in inflation results in a 2.61% increase in unemployment and a 0.06% decrease in unemployment, respectively.

The rest of the paper has the following. First, Section 2 outlines the theoretical literature review. Second, Section 3 outlines the empirical review of unemployment and inflation. Third, Section 4 discusses the methodology, including the stylized data and model specification. Fourth, Section 5 discusses descriptive statistics and empirical results. Finally, Section 6 outlines the conclusion and recommendations of the paper.

## 2. Theoretical Review

A.W. Phillips created the Phillips curve and developed it into an economic theory in which unemployment and inflation are inversely related. The Phillips curve illustrates the relationship between the level of unemployment and the rate of wage growth, with wage growth serving as a stand-in for inflation. The rate of inflation rises as unemployment decreases. As a result of the rate of inflation adjusting to new pressure of demands brought about by wage increases, there is no change in real values (Phillips 1958). The key benefit of Phillips's curve is that it can be used to solve the problem of choosing the best inflation and unemployment combination. Furthermore, it employs the impassion bend method to look at the unemployment blend. Using this theory demonstrates that lower inflation can only be achieved at the expense of higher unemployment, and higher inflation can only be achieved at the expense of lower unemployment (Phillips 1958). However, some of the key criticisms of the Phillips curve are that it disregards the part of the cash supply in creating inflation. Beneath the amount hypothesis of cash, cost levels are impacted by the amount of cash circulating within the economy. The fundamental faultfinders of the Phillips curve frequently say that it faults development for expansion and impacts arrangement without considering other causes of expansion. The Phillips curve suggests that financial development is essentially inflationary (Forder 2010).

The theory of the new Classical school of thought is based on market-clearing models, where demand and supply quickly adjust under the presumption of flexible wages and prices. Market-clearing models, according to the new Keynesian school of thought, are insufficient to explain short-term economic swings. They thus build their models on stable wages and prices, which also account for the existence of involuntary unemployment (Wilson 1986; Mankiw 2014). On the other hand, in explaining aggregate variations in terms of microeconomics underpinnings, the new Keynesian school of thought economics differs from the new classical economics. The dynamics at play are explained by the new Classical school in terms of the deliberate decisions made by people and businesses (Mankiw 2014; Mihalache and Bodislav 2019). However, modern Keynesian research shows that enterprises and households do not coordinate their decisions without incurring costs, and coordination failure is caused by coordination costs (Mihalache and Bodislav 2019). According to Okun's law, depending on the nation and period under consideration, a unit rise in cyclical unemployment is connected with two percentage points of negative growth in real GDP. There is a positive relationship between output and employment because a country's output is dependent on the labour it has employed, which also explains why there is a negative relationship between output and unemployment (Elhorst and Emili 2022).

## 3. Empirical Literature Review

Just after the adoption of inflation targeting in 2000, Hodge (2002) undertook an analysis of unemployment and inflation in SA from 1970 to 2000. It was found that a 1% increase in the unemployment rate results in a 3.61% fall in inflation when ordinary least

squares (OLS) was utilized. This provided evidence of the trade-off between inflation and unemployment, which offers the theoretical justification for the Phillips curve. Rocheteau et al. (2007) exploited the explicit microeconomic underpinnings of the general equilibrium model of unemployment and money. According to basic choice criteria, the authors predict that the implicit relationship between inflation and unemployment can either be positive or negative. They noted that the Phillips curve offers a monetary policy trade-off that can be taken advantage of in the long run, but it turns out that the Friedman rule is the best course of action.

Umaru et al. (2013) studied the relationship between unemployment and inflation from 1986 to 2010. The results show that inflation is harming unemployment. The Granger causality test found that there was no causal link between unemployment and inflation during the study period. On the other hand, if the unemployment rate and inflation rate change by 1%, the growth rate will increase by 0.21% and 0.84%, respectively. Qianyi and Fei (2013) assess the Phillips curve to establish the cause-and-effect between unemployment and inflation in China. It was found that causality is not defined in China between 1978 and 2011. It was emphasized that the Chinese economy's complexity led to the Phillips curve's inapplicability to the communist regime. Asif (2013) studied the macroeconomic determinants of unemployment from 1980 to 2009. The Johansen cointegration Granger causality test was used, and it was found that inflation insignificantly influences GDP and unemployment, and that the correlation is negative. The result of Granger causality indicates that bidirectional causality does not exist between unemployment, inflation, and economic growth. The cointegration result revealed that a long-term relationship exists among the three variables.

Amusa et al. (2013), the long-run impact of inflation in South Africa from 1960 quarter 1 to 2010 quarter 1. Within the framework of the South African economy, the long-run money superneutrality of money (LRSN)[2] hypothesis was applied. The estimation results imply that the LRSN hypothesis cannot be disproved, which implies that monetary policy in South Africa cannot be used to address the country's significant and ongoing unemployment problem. This is reasonable given that the country's persistently high unemployment rate is a result of a skills shortage. The Granger causality test and the Johansen cointegration test were utilized by Al-zeaud and Al-hosban (2015) between 1984 and 2011. It was used to examine whether there is a trade-off between unemployment and inflation in Jordan. The estimates for the elasticity of unemployment concerning inflation and the elasticity of inflation concerning unemployment were 0.02% and −0.23%, respectively. The findings indicated that there was no trade-off between the two variables throughout the study period in Jordan since there was no causal association between the inflation rate and unemployment. Vermeulen (2015) examined how inflation affects employment in SA from 1999 to 2014 using Engle Granger's error correction method. It was found that the Phillips curve hypothesis harms output, as such excessive inflation hinders the creation of jobs. Moreover, in the short term, it was found that the relationship between inflation and job growth is insignificant.

Nasr et al. (2015) investigated the asymmetric and time-varying causalities between inflation and inflation uncertainty in South Africa from 1990 month 1 to 2012 month 12. In the Gaussian Markov switching vector autoregressive (MS-VAR), there were 4 states found with transition probabilities of 0.926, 0.567, 0.821, and 0.054. The MS-VAR model shows that inflation uncertainty has a positive effect on inflation in regime 2, while inflation has a positive effect on inflation uncertainty in regime 3. In regime 4, there is bidirectional causality between inflation and inflation uncertainty, while the effects are zero in regime 1. Betul (2015), between 2001 and 2012, examined the variables influencing unemployment in BRIC nations (Brazil, Russia, India, and China). The findings showed that rising unemployment in BRIC countries is mostly caused by inflation and population expansion, while rising GDP and increased industrial production are some of the factors that contribute to falling jobless rates. Okafor et al. (2016) explored the responsiveness of unemployment to inflation in Nigeria from 1989 to 2014. They found that although there is a long-term correlation

between unemployment and inflation, there is divergence along the equilibrium phase that is fixed at a 65% annual speed of adjustment using the error correction model (ECM). The money supply and exchange rate were shown to have favourable effects on unemployment, and inflation harms it.

Bhattarai (2016) studied the trade-offs between inflation and unemployment in OECD nations from the 1990's first quarter through 2014's fourth quarter. Additionally, there were two-way Granger causal relationships between the inflation and unemployment series. On the other hand, the generalized method of moments (GMM) revealed that a 3% drop in unemployment resulted in a 1% drop in growth rate. It was noted that microeconomic structural and institutional improvements might enhance the effectiveness of pay and employment negotiations between businesses and employees, making the trade-offs between unemployment and inflation in these countries more significant and relevant. Vermeulen (2017) examined whether there is evidence of the Phillips curve in SA from 2000 to 2015. The triangle model was utilized, and it was found that a 1% increase in inflation results in a 0.936% increase in the rate of unemployment. These results outline that there is no evidence of a trade-off between inflation and the unemployment rate, thus confirming the orthodox view.

Maduku and Kaseeram (2018) investigated inflation and unemployment in SA from 1980 to 2017. Using the autoregressive distributed lag (ARDL), it was found that a 1% increase in inflation results in a 0.3245% fall in the rate of unemployment. These results are contrary to those of Vermeulen (2017). It was recommended that the SARB consider revising its objectives so that it can consider becoming involved in targeting unemployment. Donayre and Panovska (2018) studied inflation and unemployment nonlinearities from 2013 to 2015. The unemployment rate transitions above or below the two estimated thresholds, 5.03% and 7.77%, according to the threshold vector autoregression model that was used to make this determination. In addition, it changes according to whether inflation is above or below the trend rate of 0.38%. The findings indicate that during expansionary times when inflation is above its long-run trend, there is a significant negative link between wage growth and unemployment.

Korkmaz and Abdullazade (2020) investigated the causal relationship between unemployment and inflation in G6 countries from 2009 to 2017. The Granger causality test showed an indication that there is a unidirectional causality relationship between inflation rates and unemployment rates. On the other hand, it was found that because many governments adopt contractionary fiscal and monetary measures to bring inflation under control, inflation has an impact on unemployment. The value and impact of the national currency decline when inflation is strong. In a high-inflation climate, the overall demand for products and services declines as a result. Reinbold and Wen (2020) argue that there is no systemic relationship between inflation and unemployment. In the very long run, the Phillips curve is strongly positively sloped. Finally, the phase spectral analysis also shows that despite the existence of the Phillips curve at the business cycle frequency under a demand shock, the monetary policy implications are not obvious due to the unclear lead-lag relationship between inflation and unemployment. Nobrega et al. (2020) investigated the relationship between the unemployment rate and wage growth in the Brazilian economy from 2000 to 2016. The Markov-switching regression model was used, and it was found that a 1% increase in inflation will result in a 2.9198% fall in unemployment. Two regimes are found, with the first regime having a transition probability of 83.48% and the second regime having a transition probability of 79.40%. The result indicated that there is a trade-off between unemployment and wage inflation.

Papanikolaou (2020) examines the impact of unemployment and inflation from 1950 to 2017. The two-state Markov-switching dynamic model was used, and it was found that at a lower state of unemployment and a high state of unemployment, a 1% increase in inflation will result in a 0.202% increase in unemployment as well as a 0.165% fall in unemployment, respectively. It was found that structural unemployment is significant for all quintiles in states 1 and 2. In the lowest quintile, income shares increase by 0.917% in state 1 and 0.417%

in state 2. Iwasaki et al. (2021), utilizing wage inflation and unemployment, examine downwards wage rigidity (DWR). The dynamic stochastic general equilibrium (DGSE) model supported the finding that salary adjustment costs are very unequal. There is a distinct appearance of the L-shaped wage Phillips curve between wage inflation and the unemployment rate. The policy simulations raise the likelihood that the lack of wage inflation is not a long-term phenomenon and that it will likely return as the labour market continues to strengthen.

## 4. Methodology

This paper uses quantitative analysis to investigate the impact of inflation in different states of unemployment by using quarterly time series data from quarter 1 in 2008 to quarter 1 of 2022 in South Africa. The economic variables used are $\pi_t$ inflation rate $\pi_{t-1}$ is the expected inflation $une_t$ unemployment $une_t^n$ the natural rate of unemployment. The theoretical framework of the Phillips curve is used in this paper. The model that is adopted in this paper is the Markov-switching dynamic regression model (MSDRM) because it provides attractive features of transition over a set of finite states (Hansen 2000). This is important because this study seeks to investigate different states of unemployment and examine the impact of inflation in different states. Other scholars that have used the model include Nasr et al. (2015) and Nobrega et al. (2020), among others, to investigate the impact of inflation and unemployment. The theoretical framework that is adopted in this paper is that of the Phillips curve outlined in Equation (1).

$$\pi_t = \pi_{t-1} - \alpha(une_t - une_t^n) + \varepsilon_t \tag{1}$$

where $\pi_t$ is the inflation rate and $\pi_{t-1}$ is the expected inflation, which is an assumption that is adaptive expectations about inflation by economic agents based on past inflation $t-1$. The subscript $\alpha$ is the parameter measuring the response to inflation, and $une_t$ is the unemployment rate. The Phillips curve is rearranged concerning unemployment as respected in Equations (2) to (3).

$$\pi_t = \pi_{t-1} - \alpha une_t + \alpha une_t^n + \varepsilon_t \tag{2}$$

$$\alpha une_t = \pi_{t-1} - \pi_t + \alpha une_t^n + \varepsilon_t \tag{3}$$

Markov-switching dynamic regression is used for series that are believed to transition over a finite set of unobserved states, allowing the process to evolve differently in each state. The transitions occur according to a Markov process. The time of transition from one state to another and the duration between changes in the state are random (Hansen 2000). If given an economic data series denoted by $y_t$ where $t = 1, 2, \ldots, T$, is characterized by two states, such economic data series can be present in Equations (4) and (5).

$$State1 : y_t = \mu_1 + \epsilon_t \tag{4}$$

$$State2 : y_t = \mu_2 + \epsilon_t \tag{5}$$

where $\mu_1$ and $\mu_2$ are the intercept terms in $State1$ and $State2$, respectively, and $\epsilon_t$ is a white noise error with variance $\sigma^2$. The two-state model shifts in the intercept term (Hansen 2000). If the timing of switches is known, the above model can be expressed as in Equation (6).

$$y_t = s_t\mu_1 + (1 - s_t)\mu_2 + \epsilon_t \tag{6}$$

The subscript $s_t$ is 1 if the process is in state 1 and 0 otherwise. Markov-switching regression models allow the parameters to vary over the unobserved states. The MSDR model with a state-dependent intercept term is reflected in question 7.

$$y_t = s_t\mu_2 + \epsilon_t \tag{7}$$

where $\mu_{s_t}$ is the parameter of interest; $\mu_{s_t} = \mu_1$ when $s_t = 1$, and $\mu_{s_t} = \mu_2$ when $s_t = 2$. The probabilities of being in each state can be estimated with transition probabilities (Hansen 2000). One-step transition probabilities are given by $Ps_t, s_t + 1$, so for a two-state process, $p_{11}$ denotes the probability of staying in state 1 in the next period given that the process is in state 1 in the current period. Likewise, $p_{22}$ denotes the probability of staying in state 2 (Hansen 2000). The transition probabilities from one state to another can be presented in matrix 8.

$$P = \begin{pmatrix} p_{11} & p_{12} \\ p_{21} & p_{22} \end{pmatrix} \tag{8}$$

The theoretical framework outlined in Equation (3) is then extended in the Markov-switching dynamic regression, as reflected in Equation (9).

$$une_t = \begin{cases} \beta_{11} + \beta_{12}\pi_{t-1} + \beta_{13}\pi_t + \beta_{14}une_t^n + \varepsilon_{1,t} \\ \beta_{21} + \beta_{22}\pi_{t-1} + \beta_{23}\pi_t + \beta_{24}une_t^n + \varepsilon_{2,t} \end{cases} \tag{9}$$

*Stylized Data for Natural Rate of Unemployment*

The paper uses the HP filter to find the trend from the cycle of the time series (Hodrick and Prescott 1997). The trend proxies the natural rate for unemployment from the unemployment rate time series. The HP filter method is presented in Equations (10) to (11).

$$y_t = \tau_t + c_t \tag{10}$$

$$y_{t,HP}^* = \lim_{\tau_t} \left[ \sum_{t=1}^{T} (y_t - \tau_t)^2 + \lambda \sum_{t=2}^{T-1} \{ (\tau_{t+1} - \tau_t) - (\tau_t - \tau_{t-1}) \}^2 \right] \tag{11}$$

where $y_t$ is the original series, $\tau_t$ is the trend component and $c_t$ is the cyclical component of the series. where $y_t$ minimizes the sum of the squared deviation of the series $y_{t,HP}^*$ from the trend subject to the smoothing parameter $\lambda$ typical 1600 for the quarterly data, as proposed by (Hodrick and Prescott 1997). The structure of the HP filter represented in Equations (10) and (11) is reflected in Equations (12) to (13).

$$une_t = unc\_trnd_t + une\_cyc_t \tag{12}$$

$$une_t^n \equiv une\_trnd_t = une_t + une\_cyc_t \tag{13}$$

where $une_t$ unemployment $unc\_trnd_t \equiv une_t^n$ the trend of unemployment, which is rationale to be equal to the natural rate of unemployment, and $une\_cyc_t$ is the cyclical component of unemployment.

## 5. Econometric Results

Table 1 shows descriptive statistics of economic variables from 2008 quarter 1 to 2022 quarter 1. The *une* unemployment is found to have a mean of 26.52%. The *pci* inflation rate is found to have a mean of 5.88%. The *L.pci* lag variable inflation is found to have a mean value of 5.88%. Last, $une_t^n$ the natural rate of unemployment is found to have a mean value of 26.52%.

**Table 1.** Descriptive statistics of the data sourced and estimated.

| Variable | Obs | Mean | Std. Dev. | Min | Max |
|---|---|---|---|---|---|
| *une* | 57 | 26.52982 | 3.21825 | 21.5 | 35.3 |
| *pci* | 57 | 5.588772 | 2.025602 | 2.43 | 13.4 |
| *L.pci* | 56 | 5.585536 | 2.043785 | 2.43 | 13.4 |
| $hpf\_t\_une \equiv une_t^n$ | 57 | 26.52982 | 2.981687 | 22.43937 | 35.44313 |

Note: $hpf\_t\_une \equiv une_t^n$ is the natural rate of unemployment. Estimation by the author.

Table 2 estimation 1 reflects the Markov chain, dynamic regression model, from 2008 quarter 1 to 2022 quarter 1. In the first state model, estimation 1 of the *une* unemployment rate is found to have a mean of 25.55%, which is statistically significant at a 1% *p* value. The lower state of unemployment reflects that South Africa is behind in achieving the target of a 6% rate of unemployment, as outlined in the NDP (2013).

**Table 2.** Markov-switching dynamic regression.

| Estimation | 1 | 2 | 3 |
|---|---|---|---|
| Dependent variables | *une* | *une* | *une* |
| *State*1 | | | |
| *pci* | | −0.495 *** | 2.612 *** |
| | | (−4.88) | (7.93) |
| *L.pci* | | | −2.133 *** |
| | | | (−7.65) |
| $hpf\_t\_une \equiv une_t^n$ | | | 0.785 *** |
| | | | (47.24) |
| *_cons* | 25.55 *** | 28.40 *** | 2.981 |
| | (99.81) | (45.71) | (.) |
| *State*2 | | | |
| *pci* | | 1.241 * | −0.0637 |
| | | (2.42) | (−0.68) |
| *L.pci* | | | 0.0946 |
| | | | (0.91) |
| $hpf\_t\_une \equiv une_t^n$ | | | 1.107 *** |
| | | | (30.37) |
| *_cons* | 33.59 *** | 28.21 *** | −2.814 * |
| | (48.59) | (12.28) | (−2.40) |
| N | 57 | 57 | 56 |

*t statistics in parentheses* * $p < 0.05$, *** $p < 0.001$. Note: $hpf\_t\_une \equiv une_t^n$ is the natural rate of unemployment. Estimation by the author.

The lower state of unemployment rate does not reflect an economy that is labour-absorbing. This is because, in state 1, unemployment of 25.55% is lower by 0.97% than the mean unemployment rate of 26.52% in the sample of the paper. In-state 1 estimation 2, it is found that the *pci* inflation rate is statistically significant at a 1% *p* value with a negative coefficient value of 0.495. This result indicated that a 1% increase in the inflation rate will result in a 0.49% fall in the rate of unemployment at a lower unemployment rate. These results are in line with those of Nobrega et al. (2020), who found that a 1% increase in inflation will result in a 2.9198% fall in unemployment. The increase in inflation reflects an increase in the demand for goods in the market in state 1. Thus, an increase in production and hence more people are needed, which results in a fall in the rate of unemployment at a lower unemployment rate.

Table 2, state 1 estimation 3 *L.pci*, which is the lag variable that proxies expected inflation, is found to not trigger an increase in unemployment at a lower rate of unemployment. This is given in Table 2 estimation 3. The *L.pci* is found to be statistically significant at a 1% *p* value with a negative coefficient of −2.133. This result outlines that a 1% increase in the inflation expectation is associated with a 2.13% fall in the rate of unemployment in a quarter. This result suggests that at a lower unemployment rate, the labour market may be able to rationally expect the inflation rate and demand wage increase that the firm will be able to pay. This may be because a lower unemployment rate suggests that many people are employed, and there may be a union to negotiate wages, which may attract employment, resulting in a fall in the rate of unemployment. The results are in line with the rationale of more bargaining power on the employees' side at a lower state of unemployment (Mankiw 2014).

Table 2, state 2, estimation 1, reflects that the *une* unemployment rate of 33.59%, which is statistically significant at a 1% *p* value. This rate reflects that unemployed individuals may

experience financial hardship, which affects their families, relationships, and communities. When it does, consumer spending, one of an economy's main drivers of growth, declines, which can result in a recession or even a depression. Table 2, state 2, estimation 2, shows that the *pci* inflation rate is statistically significant at a 1% *p* value with a positive value of 1.241. This result indicates that a 1% increase in the inflation rate will result in a 1.241% increase in the unemployment rate. Table 2, state 2, estimation 3 *L.pci*, which is the lag variable that proxies expected inflation, is found to increase unemployment. This is given in Table 2 estimation 3. *L.pci* is found to be statistically insignificant, with a positive coefficient of 0.0946.

Figure 2 shows the filter transition probability from state 1 to state 2 of unemployment. In Figure 2, graph a represents state 1 with a mean value of 25.55%. The state is found to run from the period of 2008 quarter 1 to 2020 to quarter 2, reflecting 50 quarters. In Figure 2, graph b represents state 1 with a mean value of 33.59%. The state is found to run from 2020 quarter 2 to 2022 to quarter 1, reflecting 7 quarters. The result suggests that the most recent period shows that an increase in the unemployment rate may be associated with the prevalence of regime or state 2. This is because in 2022 quarter 1, the level of unemployment is at a rate of 34.50%.

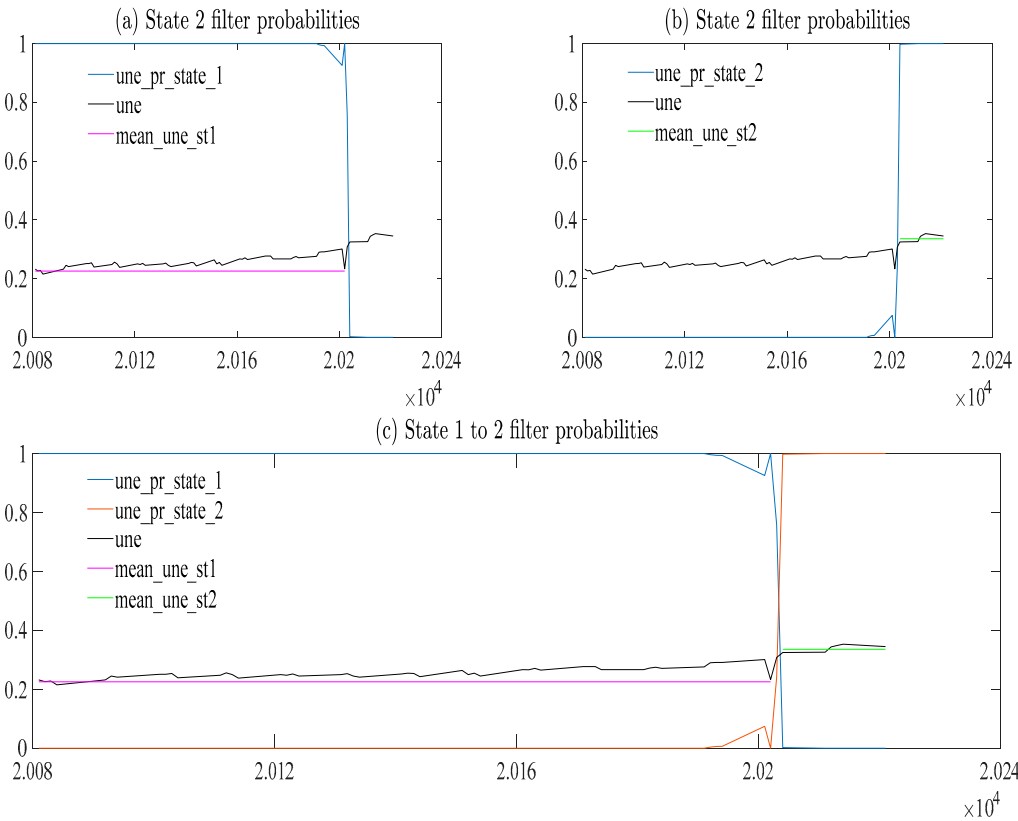

**Figure 2.** States 1 to 3 filter transition probabilities and unemployment. Note: *une_pr_state* is the transition probability of unemployment, *une* is unemployment rate data, *mean_une_*st is the mean of the unemployment rate in different states. Estimation by the author.

Table 3 reflects the matrix of transition probabilities of unemployment in different states. The first state is characterized by a mean of 25.55%. In this state, it is found to have a transition probability of 0.985121. This reflects that there is a 98.51% chance that the economy will move from state 1 and return to state 1. These results are close to those of Nasr et al. (2015), who found 0.926% transition probabilities. On the other hand, the second state is characterized by a mean of 33.59%. In this state, it is found to have a transition probability of 0.86985. This reflects that there is an 86.99% chance that the economy will

move from state 2 and return to state 2. This reflects that there is a greater chance that SA can operate in a lower state of unemployment.

**Table 3.** Matrix of transition probabilities.

| Probabilities | 1 | 2 | 3 |
| :---: | :---: | :---: | :---: |
| *p11* | 0.985054 | 0.985121 | $9.158 \times 10^3$ |
| *p12* | 0.014946 | 0.014879 | 1 |
| *p21* | 0.042506 | 0.042152 | 0.13015 |
| *p22* | 0.957495 | 0.957848 | 0.86985 |

Estimation by the author.

Table 4 shows the expected duration of each state. When the economy is in state 1, estimation 2 is found to run for 67 quarters. When the economy is in state 2, estimation 3 is found to run for 7 quarters. This reflects that the SA economy has the potential to operate for a long time in a lower state of unemployment.

**Table 4.** Expected duration.

| State | 1 | 2 | 3 |
| :---: | :---: | :---: | :---: |
| State1 | 66.90567 | 67.20864 | 1 |
| State2 | 23.52637 | 23.72362 | 7.683427 |

Estimation by the author.

## 6. Conclusions

This paper investigated the impact of inflation in different states of unemployment: evidence with the Phillips curve in South Africa. The key questions of this paper are as follows: what is the impact of inflation in different states of unemployment? How long will the unemployment rate be at a higher rate of unemployment and a lower rate of unemployment? What is the probability of transition to different states of unemployment? Given these questions, the purpose of this paper is to investigate the impact of inflation in different states of unemployment with evidence of the Phillips curve in South Africa. The contribution of this paper is to examine the impact of inflation in different states of unemployment. The paper employs Markov-switching dynamic regression (MSDR) using time-series data from 2008 quarter 1 to 2022 quarter 1. It was found that there are 2 states of unemployment with mean rates of 25.55% and 33.59%. These states are expected to run for 67 and 7 quarters, respectively. Moreover, there is a 98.51% and 86.99% chance of a move and then returning to the same state in both states 1 and 2, respectively. In states 1 and 2, a 1% increase in inflation results in a 2.61% increase in unemployment and a 0.06% decrease in unemployment, respectively. It is important to note that these results do not reflect a direct cause-and-effect relationship between unemployment. Thus, it is recommended that future studies use the Granger causality test. The limitation of the paper is that other economic variables are not considered, but they influence unemployment, which includes education, population, work experience, and skills, among others.

It is recommended that inflation be kept at a lower rate when the economy is operating at an unemployment rate of 25.55% or state 1. There is a need to explore the possibility of increasing the South African Reserve Bank mandate to include lowering unemployment. This is because unemployment is found to be responsive in state 1 when there is a change in inflation. State 2 has a high rate of unemployment with a mean of 33.59%, and inflation and unemployment have a negative relationship. This result suggests that there are trade-offs between unemployment and inflation that pose a dilemma for South African policymakers since to reduce unemployment, the inflation rate in the economy tends to rise. The government needs to invest in labour-absorbing sectors of the economy, such as the agricultural sector and manufacturing. With this, it may be expected that there will be an increase in the supply of products. The increased supply will reduce prices and increase

employment generation. It is recommended that the government increase the channels of employment opportunities. Moreover, there is a need to reduce the cost of applying for a job. This the government can achieve by putting all jobs online for application. Moreover, there is a need for data collection of people's skills and qualifications in the effort to create a job data bank. Its recommendations for future study take into account income disparity in the tool analyses in relation to finding the impact it has on inflation.

**Funding:** This research received no external funding.

**Data Availability Statement:** The data are available at https://drive.google.com/drive/folders/1U57TzqDG-fM6YqvKgjT7F69CspGUfu0H?usp=share_link (accessed date: 1 November 2022).

**Acknowledgments:** I would like to acknowledge the University of Free State Economic Department for giving me time to do research and reducing my workload as an nGAP lecturer.

**Conflicts of Interest:** There is no conflict of interest in the paper.

## Notes

[1]   Unemployment is a situation of being out of work or needing a job and searching for it continuously in the last four weeks or remaining unemployed at age 16 or above but available to join work in the next two weeks.

[2]   Reflect the changes in the growth rate of money volume, no changes in the real production level have been observed, and so the hypothesis of superneutrality of money is confirmed.

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
