# Peer review of "Impact of Inflation in Different States of Unemployment: Evidence with the Phillips Curve in South Africa from 2008 to 2022"

_economies, doi:10.3390/economies11010029_

Round 1

Reviewer 1 Report

The paper is interesting and represents a high scientific level. The arrangement of the considerations is logical, and the authors try to embed the research in the scientific economic literature. The conclusions are substantial and consistent with the research.

The merit of the authors is the critical approach to the concept of the Philips curve in terms of cause-and-effect relationships between inflation and unemployment. The authors recognize the limitations of the study and correctly interpret the results obtained. However, I suggest that in the "Conclusion" section it should be more precisely indicated that demonstrating the relationship between unemployment and inflation does not have to mean direct cause-and-effect relationships, especially if recommendations are to be formulated responsibly. The issue under study is much more complicated. With regard to the constraints on labor market segmentation, it is worth considering the use of the equilibrium unemployment estimation method developed by the Center for Economic Policy Research, published in 1995 in the “Unemployment: Choices for Europe” report in subsequent studies.

Author Response

Dear Editor 

Kindly find the attached report on how I have addressed the comments of the reviews. I have also addressed the grammar, and spelling, to ensure that the economic ideas are communicated clearly as well as with the academic follow. All the changes done in the document are on-track changes. 

Review one 

The paper is interesting and represents a high scientific level. The arrangement of the considerations is logical, and the authors try to embed the research in the scientific economic literature. The conclusions are substantial and consistent with the research.

-Noted 

The merit of the authors is the critical approach to the concept of the Philips curve in terms of cause-and-effect relationships between inflation and unemployment. The authors recognize the limitations of the study and correctly interpret the results obtained. 

-Noted 

However, I suggest that in the "Conclusion" section it should be more precisely indicated that demonstrating the relationship between unemployment and inflation does not have to mean direct cause-and-effect relationships, especially if recommendations are to be formulated responsibly. 

-This is addressed in the document now. It is outlined under the conclusion that “It is important to note that these result does not reflect a direct cause-and-effect relationships relationship between unemployment. As such it is recommended that future studies used the granger causality test”

The issue under study is much more complicated. 

-As acknowledged by the reviewer some of the issues are complicated. This may be down to the econometric technic and theoretical framework. However, with the economic question of interest in the paper the Markov-switching dynamic regression (MSDR) is perfect compared to the VAR, EVC, and OLS which are limed to the long-run estimation.

With regard to the constraints on labor market segmentation, it is worth considering the use of the equilibrium unemployment estimation method developed by the Center for Economic Policy Research, published in 1995 in the “Unemployment: Choices for Europe” report in subsequent studies

-The HP filter is to find the natural rate of unemployment. The equilibrium unemployment estimation method was not used because there was no data available in the effort to estimate the equilibrium unemployment estimation method. However, even I the data was available the HP filter would be a more preferred method. This is for comparison with other studies that also used the HP filter. As such it is proposed that the HP filter that is used in the paper be kept and the equilibrium unemployment estimation method needs to be put aside. 
Thanks 
The Author

Reviewer 2 Report

The topic proposed by the author, although it is not new, manages to bring into discussion problems on which, probably, there will never be a total agreement among researchers. Intuited, demonstrated, disproved, the Phillips curve remains an exciting topic every time.

The methodology is applicable, the conclusions are natural, we have no other comments. Regarding the bibliographic sources, in our opinion, the aspects related to competitiveness should not be neglected in the context of wage policies that can have an impact both on inflation and especially on unemployment. Regarding the literature review, a recent research was carried out by Avram, A.; Benvenuto, M.; Avram, C.D.; Gravili, G. Assuring SME’s Sustainable Competitiveness in the Digital Era: A Labor Policy between Guaranteed Minimum Wage and ICT Skill Mismatch. Sustainability 2019, 11, 2918. https://doi.org/10.3390/su11102918 with results relevant to your paper. Thus, I strongly recommend to cite the following paper.

A revision of the text is necessary because there are still mistakes that can be disturbing (lines 145-146 UMARU, DONGA et al. (2013), studied the relationship between unemployment and unemployment from 1986 to 2010).

Author Response

Dear Editor

Kindly find the attached report on how I have addressed the comments of the reviews. I have also addressed the grammar, and spelling, to ensure that the economic ideas are communicated clearly as well as with the academic follow. All the changes done in the document are on track changes.

Review two

The topic proposed by the author, although it is not new, manages to bring into discussion problems on which, probably, there will never be a total agreement among researchers. Intuited, demonstrated, disproved, the Phillips curve remains an exciting topic every time.

  • Noted

The methodology is applicable, the conclusions are natural, we have no other comments.

  • Noted

Regarding the bibliographic sources, in our opinion, the aspects related to competitiveness should not be neglected in the context of wage policies that can have an impact both on inflation and especially on unemployment. Regarding the literature review, recent research was carried out by Avram, A.; Benvenuto, M.; Avram, C.D.; Gravili, G. Assuring SME’s Sustainable Competitiveness in the Digital Era: A Labor Policy between Guaranteed Minimum Wage and ICT Skill Mismatch. Sustainability 2019, 11, 2918. https://doi.org/10.3390/su11102918 with results relevant to your paper. Thus, I strongly recommend to cite the following paper.

  • I agree with the review that wages play an important role in the level of unemployment. However, the available data on wages in South Africa is no yearly data and it reflects the government employees, not another sector which can be a disadvantage. There is an econometrics technic that can change data from yearly to quality however that needs a lot of justification which can be beyond the scope of this paper. As far as the scope of the paper the macroeconomic variable of unemployment is sufficient. Moreover, given the economic questions, the paper needs to focus on the relationship between inflation and unemployment. Remember again we are looking at things in the context of the Phillips curve. I believe the inclusion of wages will broaden the scope of the paper and will require the question of what the impact of wages on unemployment and inflation is which is not the focus of the paper. This then suggests that the wage, inflation, and unemployment can be a different paper considering theoretical framework, data alignment, economic question, and econometric model to handle the scope of the paper.
  • It is in this regard that I propose not to include wage in the paper that is prosed by the review. This paper does not talk about the relationship between inflation and unemployment. However, it talks about wages which may have an effect on unemployment. Nonetheless, because of the limitation and the boarding of the scope of the paper as outlined, I wish to not include the economic variable of wage and not cite the paper.
  • I will acknowledge the importance of wages in the conclusion and recommend that future studies need to look at the aspect of wages, inflation, and unemployment.

A revision of the text is necessary because there are still mistakes that can be disturbing (lines 145-146 UMARU, DONGA et al. (2013), studied the relationship between unemployment and unemployment from 1986 to 2010).

  • This is addressed you can see the track change in the whole document in the effort to address spelling, grammar and sentence struct to follow well and better community the economic ideas.

Looking forward to your positive response.

Thanks

The Author 
